# Response of Moso Bamboo Growth and Soil Nutrient Content to Strip Cutting

**Xiao Zhou** [1,2], **Fengying Guan** [1,2,*], **Xuan Zhang** [1,2], **Chengji Li** [1,2] **and Yang Zhou** [1,2]

1   International Center for Bamboo and Rattan, Key Laboratory of National Forestry and Grassland Administration, Beijing 100102, China

2   National Location Observation and Research Station of the Bamboo Forest Ecosystem in Yixing, National Forestry and Grassland Administration, Yixing 214200, China

*   Correspondence: guanfy@icbr.ac.cn; Tel.: +86-10-84789808

**Abstract:** Moso bamboo (*Phyllostachys edulis*) is a critical forest resource in subtropical China, and reasonable cutting management of moso bamboo forests is essential for improving the productivity of bamboo forests, increasing the income of farmers, and improving the ecological environment. Therefore, we set up sample plots with different cutting widths at the Yixing Forest Farm in Jiangsu Province in December 2017. Moso bamboo growth surveys and soil sampling were conducted in May 2018 to study the effects of different cutting widths on the growth and nutrient content of moso bamboo forests. Our results indicate that strip cutting had significant effects on degraded bamboo shoots, the number of new bamboos, and their ratios. Soil elements showed surface aggregation, and cutting increased the soil nutrient content. Principal component analysis showed that stand characteristics (diameter at breast height and number of new bamboo shoots) were positively associated with total phosphorus and available phosphorus but negatively correlated with available potassium, total potassium, and soil organic carbon. A cutting width of 8 m resulted in rich nutrient content, which is suitable for bamboo cultivation. These results will provide theoretical guidance for the formulation of scientific and reasonable strip cutting methods for moso bamboo forests.

**Keywords:** soil sample; soil physicochemical analysis; strip cutting; moso bamboo growth; soil depth

## 1. Introduction

Bamboo is a non-woody plant that is common in forest ecosystems worldwide [1–4] and is an important component of many forest ecosystems [5]. China has the richest bamboo resources in the world in terms of number of species (>500 species in 39 genera) and area. The area of bamboo forests has steadily increased and is currently at approximately 6.41 million hectares (ha), accounting for approximately 2.94% of the forest area, with 72.96% of that area being occupied by moso bamboo forests (data from the Ninth National Forest Resource Inventory Report).

Moso bamboo (*Phyllostachys edulis*) is an economically important bamboo species. The annual biomass of bamboo stalk in moso bamboo forests can generally reach 8.25–9 t/ha with an input–output ratio of 1:2–1:4, which can increase the economic benefit two to three times more than with the management of general timber forest species [6]. Sustainable cutting management of moso bamboo forests is critical for improving their productivity, increasing the income of farmers, and improving the ecological environment. Previous studies have emphasized the economic value of bamboo, such as providing non-woody materials and food for humans [7,8]. Since the signing of the Kyoto Protocol, woody plants have attracted considerable attention [4,6,9] and include bamboo forest carbon capacity [9,10], management effects [11,12], and bamboo encroachment effects [13,14].

Owing to the growth characteristics of moso bamboo, specific management practices have been widely applied in the last few decades to increase its growth and thus achieve higher economic returns [15,16], including inorganic fertilizer application, cutting, tillage,

and regular removal of understory vegetation. These cutting methods are selective; however, previous studies have indicated that the long-term application of these practices may have negative ecological consequences, such as soil erosion and nutrient leaching [17], increases in soil $CO_2$ emissions [15], acceleration of soil organic C mineralization, and decrease in soil organic C storage [18]. These management methods will affect the ecosystem, and the continuous increase in labor and time costs coupled with manual selective cutting cannot meet the needs of bamboo forest cultivation and industry development [19]. Ultimately, these will have serious effects on the sustainable management of plantations.

Strip cutting of bamboo forests resolves the issue of labor cost restriction of traditional selective cutting management, changes the labor-intensive cultivation mode, and allows mechanized management of bamboo forests [15]. Thus, this method has become the main focus and direction of research in bamboo cultivation technology.

Many researchers choose the optimal cutting width by evaluating various indicators of different cutting widths [20–23]. Su et al. [20] selected the optimal width by comparing the stand characteristics of new bamboo after cutting. Zeng et al. [21,22] compared different soil nutrients, and Wang [23] compared the selection of microbial changes in different cutting widths. Currently, research on strip cutting of moso bamboo forests is still preliminary and focuses on the number of ground bamboo [20], understory shrub and grass, and bamboo forest biomass [21] after cutting. Previous studies have used a single aspect of comparison to select the optimal cutting width. However, there are few reports on the changes in soil nutrients and stand characteristics in different cutting zones after the strip cutting of moso bamboo forests [22].

Soil is an important medium for plant survival and is a key source of water, heat, and fertilizers for plant growth. Many studies have shown that forest soil quality is closely related to vegetation growth [20–22]. Different soil management types may result in differences in soil nutrient content and affect bamboo growth [11]. It is therefore necessary to understand these effects.

The effects of different cutting widths on soil nutrients and stand characteristics of bamboo forests are unclear. In this study, strip cutting with different cutting widths was performed using the unrecovered sample plot as the control. It is assumed that banded cutting can adjust the competitive relationship between moso bamboo individuals, provide a suitable environment for moso bamboo growth, and thus promote growth and development. This study determined (1) whether there are differences in soil nutrient content and stand characteristics at different cutting widths, (2) the effects of soil nutrient factors on stand characteristics after harvesting at different cutting widths, and (3) the optimal cutting width by correlation analysis. These findings will provide a scientific basis for the rational management of banded cutting of moso bamboo forests.

## 2. Materials and Methods

### 2.1. Study Site

This study was conducted in Yixing City, a state-owned forest farm (119°41′–119°44′ E, 31°13′–31°15′ N) in Jiangsu Province, China (Figure 1). This region is dominated by low mountains and hilly terrain, and the soil type is yellow clay, according to the classification and codes for Chinese soil (GB/T 17296-2009). The area is characterized by a subtropical monsoon climate with an average annual precipitation and temperature of 1167 mm and 15.7 °C, respectively. The annual sunshine duration is 1807.5 h, and the annual evaporation is 886.8 mm. The terrain is dominated by plains and hills, which are the areas with the most abundant bamboo forests in Jiangsu Province. The vegetation type in the study area was a pure moso bamboo forest.

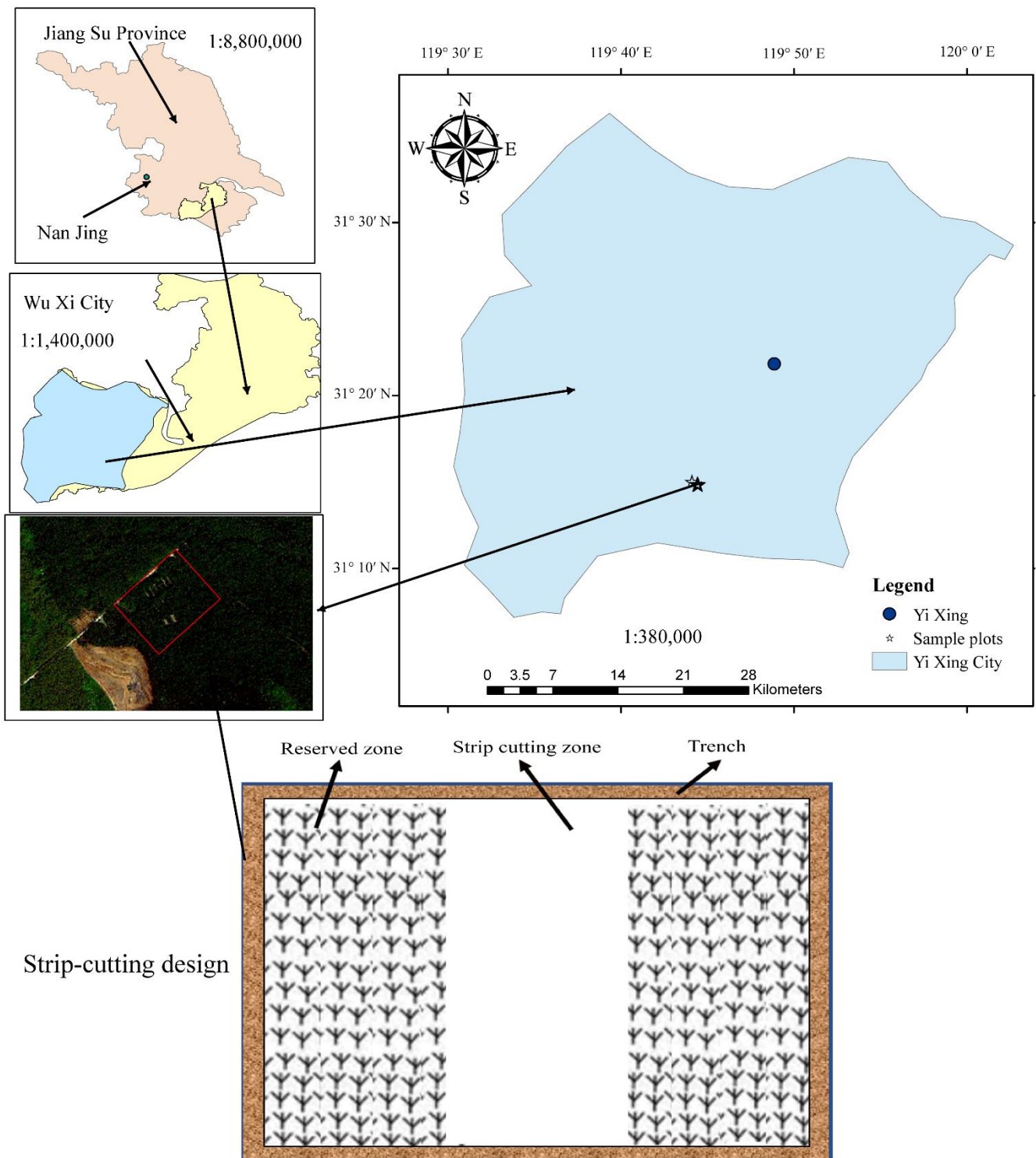

**Figure 1.** Study area showing the sample plot locations and strip-cutting design.

## 2.2. Experimental Design and Measurement

Strip cutting involves cutting down all the trees in a plot and removing the whole plant from the experimental site. A complete rotation system reserved two belts between each cut belt, cutting one reserved belt after the cutting-belt restoration, and then cutting another reserved belt after the reserve-belt cut restoration. The function of the reserved plots was to provide nutrients to strip-cut belts through physiological integration. Each cut plot had two retention plots to provide nutrients for the recovery period. No management practices were

implemented in the harvest and reserve plots during the restoration period. In December 2017, we selected a pure moso bamboo forest with a tree density of 3500 plants/ha, a diameter at breast height (DBH) of approximately 9.8 cm, and the age structure of grades I, II, and III bamboo was 3:4:3. A 20 m × 20 m moso bamboo forest was used as the control plot (CK) with a cutting width of 0 m (CK), 3 m (M3), 5 m (M5), 8 m (M8), and 12 m (M12) and a length of 20 m. Clear cutting was performed in a quadrat of the moso bamboo forest. Each treatment was repeated three times, for a total of 15 sample plots. Four trenches (50 cm wide × 50 cm deep) were excavated around the plot to cut off rhizomes from the influence of long-distance nutrient transport. The cutting design drawing of the strip-cutting sample plot and satellite aerial photograph are shown in Figure 1, respectively. In addition, the slope of the sample plot selected in this study was gentle, and the slope direction had no influence on the experimental process and results.

*2.3. Soil Sampling and Soil Physicochemical Analysis*

Soil samples were collected from 15 plots in May 2018. Within each plot, soil samples were taken at depths of 0–10, 10–20, 20–30, and 30–50 cm from five randomly selected points. Five soil cores from the same layers were placed in a clean plastic bucket, mixed thoroughly to form a composite sample, and brought to the laboratory (60 in total). After removing all visible roots and plant fragments (>2 mm diameter), the soil cores were air-dried at room temperature and ground to pass through a 2-mm or 0.25-mm sieve for chemical analysis.

The soil pH was determined using a soil-to-water ratio of 1:2 and a pH meter. The soil organic carbon (SOC) concentration was determined by wet digestion with 133 mmol $L^{-1}$ $K_2Cr_2O_7$ and concentrated $H_2SO_4$ at 170–180 °C. Total nitrogen content was determined using the Kjeldahl method. The total phosphorus level was determined using sodium hydroxide melting molybdenum antimony anti-colorimetry. Total potassium content was determined by sodium hydroxide melting atomic absorption spectrophotometry, while alkali-hydrolyzed nitrogen was determined by the alkali hydrolysis diffusion method. Available phosphorus was extracted using hydrochloric acid ammonium fluoride and measured using molybdenum antimony anti-colorimetry, while available potassium was extracted using ammonium acetate and measured using atomic absorption spectrophotometry. Soil moisture was evaluated by oven drying a subsample of soil at 105 °C for 24 h. After determining the soil moisture content of the bulk density samples, we calculated the bulk density based on the volume and total oven-dry weight of the soil within each soil core [24]. SOC stocks were calculated using the following equation:

$$CS_{soil} = \sum_{i=1}^{n} BD_i * Cconc_i * D_i * 0.1 \tag{1}$$

where $CS_{soil}$ represents the soil carbon storage (Mg C/ha), $BD_i$ is the soil bulk density in layer i (Mg/m$^3$), $C_{conci}$ is the SOC concentration in layer i (g/kg), $D_i$ is the thickness of the soil layer (cm), and i is the soil layer number.

*2.4. Aboveground Biomass*

Aboveground biomass was estimated from DBH, and data was measured from each culm in the experimental plot using allometric equations derived near the study plot. During the fieldwork, the diameter of the moso bamboo at DBH and age (du) were measured; 1–2-year-old bamboo or new birth bamboo were referred to as 1 du, 3–4-year-old bamboo as 2 du, and 5–6-year-old bamboo as 3 du. Fifteen sample plots were collected, and their spatial distributions are illustrated in Figure 1. In this study, the model, including age proposed by Zhou, was used for biomass calculation [25], and 66 moso bamboos were harvested from a Yixing state-owned forest farm to correct the initial value of the model [4].

The corrected model was then used to calculate the biomass per plant, and the formula is as follows (Equation (2)):

$$M(D, A) = 0.7932D^{1.8282} \left( \frac{0.9964A}{A + 0.0005} \right)^{213.9988} \tag{2}$$

where *M*, *D*, and *A* represent the AGB (aboveground biomass), DBH (cm), and du, respectively. For each plot, the AGB was the sum of all individual moso bamboo AGB within the plot.

### 2.5. Statistical Analyses

One-way analysis of variance (ANOVA) was used to test whether the soil nutrient content and new bamboo characteristics differed among the five treatment plots. Assumptions of normality and homogeneous variance were examined using the Shapiro–Wilk test and Levene's test, respectively. Means were separated using the least significant difference test, and statistical significance was set at $p < 0.05$. A principal component analysis (PCA) was used to examine the associations between functional traits and stand and soil characteristics. All statistical analyses were performed using R (version 4.1.0) and SPSS statistics software (version 17.0). PCA was calculated using the 'psych' package. Graphs were drawn using the 'ggplot2' package.

## 3. Results
### 3.1. Effects of Strip Cutting on the Growth of Moso Bamboo Forest

The characteristics of bamboo forests with different cutting widths are shown in Table 1. Strip cutting significantly reduced the number of degraded bamboo shoots ($p < 0.05$). With an increase in the cutting zone width, trees with degraded bamboo shoots first decreased and then increased, reaching a minimum at M8.

**Table 1.** Stand characteristics for moso bamboo in strip cutting (M3, M5, M8, M12, and M15) or unharvested (CK) plots.

| Treatment | DBH | DBS | NB | DBS: NB |
|---|---|---|---|---|
| M3 | 9.40 ± 0.67 a | 300 ± 124.72 c | 1433 ± 543.65 b | 0.23 ± 0.12 b |
| M5 | 8.22 ± 0.67 ab | 316.67 ± 195.08 c | 2366.67 ± 815.82 ab | 0.13 ± 0.07 b |
| M8 | 8.28 ± 1.79 ab | 265.63 ± 102.46 c | 2234.38 ± 628.70 ab | 0.11 ± 0.02 b |
| M12 | 7.79 ± 0.11 b | 861.11 ± 109.36 b | 2402.78 ± 437.44 a | 0.37 ± 0.10 b |
| M15 | 7.77 ± 0.22 b | 1700 ± 288.03 a | 2788.89 ± 245.45 a | 0.61 ± 0.11 b |
| CK | 9.71 ± 0.22 a | 693.75 ± 81.73 b | 393.75 ± 139.61 c | 2.02 ± 0.80 a |

Notes: Data are presented as the mean ± SD. Different lowercase letters (a, b, c) indicate significant differences among cutting widths ($p < 0.05$). The same below. DBS (degraded bamboo shoot); NB (number of new bamboo shoots).

In addition, the number of new bamboos with different cutting widths was significantly higher than that of the control group (CK) ($p < 0.05$). Although there were significant differences among M3, M12, and M15 ($p < 0.05$), differences among other groups were not significant ($p > 0.05$). With an increase in cutting width, the number of new bamboo show a trend of increasing and decreasing repeatedly.

The DBH of new bamboo in each cutting zone decreases from CK, M3, M8, M5, M15, and M12; only M3, M12, and M15 had significant differences ($p < 0.05$), and the other had no significant difference ($p > 0.05$).

For this ratio (degraded bamboo shoot to number of new bamboo), a significant difference was observed between the control group and the different cutting widths.

### 3.2. Effects of Strip Cutting on Soil Elements of Moso Bamboo

At different soil depths, SOC, total nitrogen (TN), hydrolytic nitrogen (HN), available phosphorus (AP), and available potassium (AK) showed significant differences ($p < 0.05$)

with these elements being the most abundant in the topsoil. Although there were no significant differences in total phosphorus (TP) and total potassium (TK) ($p > 0.05$), they were mainly present at the top of the soil. The soil pH became more acidic with increasing soil layer depth (Table 2).

For different cutting widths, the variation in nutrient content in each soil layer was inconsistent. Except for TN and TP, the contents of the other nutrients in each soil layer did not differ significantly among cutting widths ($p > 0.05$) (Table 2).

For different cutting widths, the TN and TP at 5 m were significantly different from those of the others and less than those of the others ($p < 0.05$) and reached the maximum in M8, M15, and M15, respectively (Table 2).

With an increase in cutting width, SOC, TN (total nitrogen), TP (total phosphorus), TK, HN, and AP in each soil layer first decreased and then increased. The SOC, TP, HN, and AP contents were the lowest when the cutting width was M5, except for HN, which was the lowest when the cutting widths were M12 and M15. The cutting widths with the highest contents were M12, M15, and M8. With increased cutting width, the content of available potassium first decreased, then increased, and finally decreased, reaching a minimum in M15 and a maximum in M8. There was no significant difference in pH among soil layers, but the pH reached a maximum when the cutting zone was M12 and a minimum in M3 and M5 ($p > 0.05$) (Table 2).

One-way ANOVA showed that the cutting width had a significant effect on C storage changes. As shown in Figure 2, SOC storage fluctuated with the increase in cutting width.

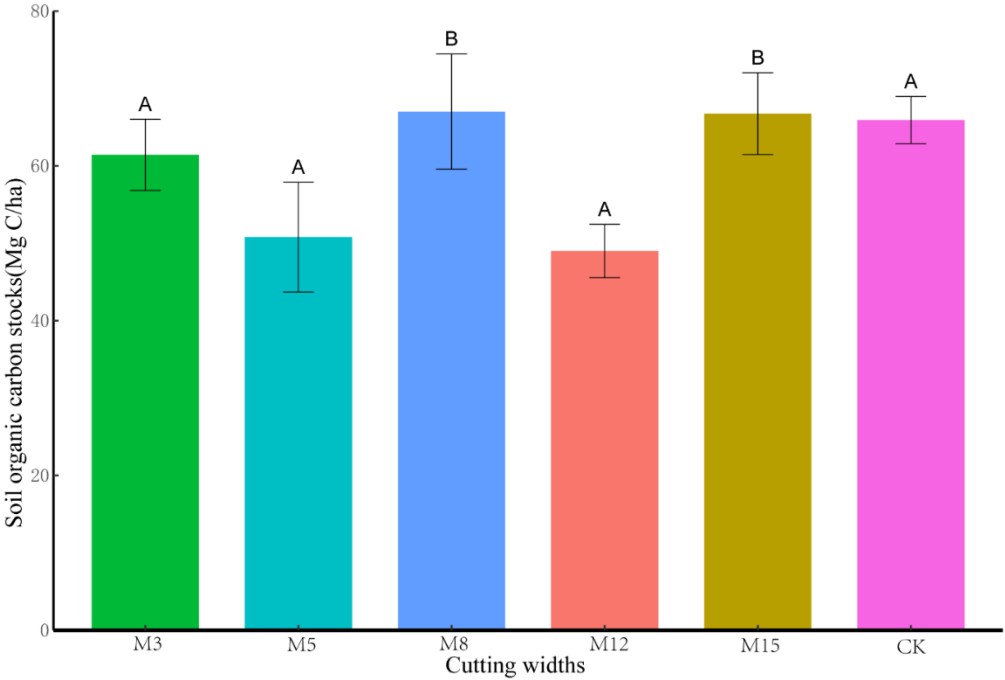

**Figure 2.** Soil organic carbon content for moso bamboo in strip cutting (M3, M5, M8, M12, and M15) or unharvested (CK) plots. Different uppercase letters (A, B) indicate significant dif-ferences among cutting widths ($p < 0.05$).

Compared with the CK group, M5 and M8 were significantly different from other groups. After cutting, the digging isolation ditch can alter SOC storage. At M8 and M12, the SOC storage was greater than that of the CK group, but the difference was not significant ($p > 0.05$).

**Table 2.** Soil characteristics for moso bamboo in strip cutting (M3, M5, M8, M12, and M15) or unharvested (CK) plots.

| Nutrient Element | Soil Depth/cm | PH | SOC (g/kg) | TN(g/kg) | TP(g/kg) | TK (g/kg) | HN (mg/kg) | AP (mg/kg) | AK (mg/kg) |
|---|---|---|---|---|---|---|---|---|---|
| M3 | 0–10 | 5.33 ± 0.05 Aab | 35.98 ± 2.58 Aa | 1.84 ± 0.15 Aab | 0.27 ± 0.01 Ac | 10.12 ± 0.82 Aa | 164.15 ± 16.03 Aab | 1.38 ± 0.42 Aab | 66.77 ± 7.60 Aa |
|  | 10–20 | 5.25 ± 0.12 ABc | 24.43 ± 2.50 Bbc | 1.30 ± 0.13 Bbc | 0.24 ± 0.01 Bbc | 10.04 ± 0.51 Aa | 106.22 ± 14.18 Bbc | 0.73 ± 0.18 Bab | 49.00 ± 5.57 Bca |
|  | 20–30 | 5.15 ± 0.20 Bb | 18.75 ± 3.71 Ca | 1.03 ± 0.18 Ca | 0.23 ± 0.01 BCcd | 10.08 ± 0.53 Aa | 91.65 ± 22.77 Ba | 0.59 ± 0.31 Bcab | 51.32 ± 10.21 Ba |
|  | 30–50 | 4.99 ± 0.11 Cb | 11.84 ± 1.56 Da | 0.78 ± 0.09 Dab | 0.23 ± 0.01 Cbc | 10.65 ± 0.62 Abc | 55.47 ± 8.38 Cab | 0.38 ± 0.14 Cb | 42.05 ± 5.46 Ca |
| M5 | 0–10 | 5.32 ± 0.11 Ab | 29.03 ± 7.29 Ab | 1.54 ± 0.33 Ab | 0.24 ± 0.01 Ad | 9.85 ± 0.44 Ba | 140.13 ± 20.02 Ab | 1.02 ± 0.36 Ab | 62.60 ± 7.12 Aa |
|  | 10–20 | 5.35 ± 0.11 Abc | 20.52 ± 1.32 Bc | 1.14 ± 0.07 Bc | 0.22 ± 0.02 Abc | 9.96 ± 0.64 Bab | 101.41 ± 12.11 Bc | 0.59 ± 0.09 Bb | 45.55 ± 5.51 Ba |
|  | 20–30 | 5.13 ± 0.10 Bb | 12.69 ± 1.85 Cb | 0.82 ± 0.12 Ca | 0.20 ± 0.01 Be | 10.40 ± 0.52 Aba | 67.96 ± 14.24 Ca | 0.42 ± 0.14 Bb | 43.59 ± 5.04 Bab |
|  | 30–50 | 4.97 ± 0.10 Cb | 8.85 ± 1.09 Cb | 0.66 ± 0.06 Cb | 0.21 ± 0.01 Bc | 10.89 ± 0.68 Aab | 45.63 ± 6.27 Db | 0.32 ± 0.12 Bb | 44.03 ± 4.40 Ba |
| M8 | 0–10 | 5.39 ± 0.07 Aab | 37.31 ± 2.48 Aa | 1.88 ± 0.03 Aa | 0.28 ± 0.01 Abc | 9.52 ± 0.94 Aa | 177.19 ± 9.53 Aa | 1.24 ± 0.21 Aab | 66.11 ± 3.85 Aa |
|  | 10–20 | 5.39 ± 0.15 Abc | 26.92 ± 2.73 Bab | 1.39 ± 0.09 Bab | 0.24 ± 0.01 Abc | 9.80 ± 0.45 Aab | 126.06 ± 14.24 Bab | 0.73 ± 0.13 Bab | 47.65 ± 5.30 Ba |
|  | 20–30 | 5.19 ± 0.11 Abb | 17.85 ± 2.88 Ca | 1.01 ± 0.18 Ca | 0.23 ± 0.01 Ad | 10.37 ± 0.43 Aa | 94.50 ± 21.01 Ca | 0.44 ± 0.32 BCb | 52.45 ± 11.51 Ba |
|  | 30–50 | 5.09 ± 0.11 Bb | 12.63 ± 2.06 Da | 0.94 ± 0.21 Ca | 0.27 ± 0.06 Aab | 10.06 ± 0.98 Abc | 70.07 ± 18.27 Ca | 0.31 ± 0.11 Cb | 44.79 ± 6.14 Ba |
| M12 | 0–10 | 5.49 ± 0.10 Aa | 38.97 ± 3.10 Aa | 2.08 ± 0.20 Aa | 0.29 ± 0.01 Ab | 8.10 ± 0.19 Bb | 189.86 ± 29.29 Aa | 1.82 ± 0.38 Aa | 58.93 ± 2.91 Aa |
|  | 10–20 | 5.62 ± 0.12 Aa | 29.88 ± 3.17 Ba | 1.53 ± 0.15 Ba | 0.26 ± 0.01 Bb | 10.21 ± 0.47 Aa | 144.99 ± 5.72 Ba | 1.27 ± 0.31 Aba | 48.61 ± 4.16 Ba |
|  | 20–30 | 5.56 ± 0.06 Aa | 15.34 ± 0.96 Cab | 0.93 ± 0.05 Ca | 0.26 ± 0.01 Bb | 10.73 ± 0.93 Aa | 72.49 ± 5.59 Ca | 0.78 ± 0.16 Bcab | 39.54 ± 2.33 Cab |
|  | 30–50 | 5.30 ± 0.04 Ba | 10.14 ± 0.61 Ca | 0.74 ± 0.03 Cab | 0.25 ± 0.02 Babc | 11.06 ± 1.01 Aa | 50.27 ± 4.40 Cb | 0.42 ± 0.16 Cab | 37.65 ± 3.14 Ca |
| M15 | 0–10 | 5.33 ± 0.13 Abab | 40.97 ± 1.36 Aa | 2.14 ± 0.03 Aa | 0.36 ± 0.02 Aa | 9.35 ± 0.66 Aa | 193.53 ± 12.69 Aa | 1.70 ± 0.15 Aa | 55.83 ± 4.95 Aa |
|  | 10–20 | 5.51 ± 0.05 Aab | 28.34 ± 3.38 Bab | 1.48 ± 0.16 Bab | 0.32 ± 0.03 Aba | 9.41 ± 0.72 Aab | 132.69 ± 9.17 Ba | 1.05 ± 0.04 Bab | 43.16 ± 0.87 Ba |
|  | 20–30 | 5.44 ± 0.04 Aba | 20.12 ± 2.50 Ca | 1.15 ± 0.16 Ca | 0.29 ± 0.02 Ba | 9.68 ± 0.66 Aa | 101.19 ± 16.63 Bca | 0.97 ± 0.13 Ba | 36.50 ± 2.93 Bb |
|  | 30–50 | 5.27 ± 0.07 Ba | 13.58 ± 1.76 Da | 0.85 ± 0.15 Cab | 0.29 ± 0.02 Ba | 9.59 ± 0.54 Ac | 72.71 ± 15.72 Ca | 0.62 ± 0.02 Ca | 28.80 ± 0.78 Cb |
| CK | 0–10 | 5.46 ± 0.06 Aab | 38.52 ± 2.35 Aa | 2.06 ± 0.11 Aa | 0.28 ± 0.01 Abc | 9.42 ± 0.86 Aa | 186.41 ± 20.07 Aa | 1.30 ± 0.40 Aab | 63.35 ± 8.53 Aa |
|  | 10–20 | 5.45 ± 0.04 Aabc | 26.01 ± 0.63 Bab | 1.43 ± 0.06 Bab | 0.25 ± 0.01 Bb | 8.99 ± 0.78 Ab | 133.98 ± 20.71 Ba | 0.65 ± 0.01 Bab | 45.70 ± 5.04 Ba |
|  | 20–30 | 5.09 ± 0.01 Bb | 17.21 ± 2.25 Cab | 0.95 ± 0.14 Ca | 0.25 ± 0.02 Bbc | 10.21 ± 0.66 Aa | 80.26 ± 17.48 Ca | 0.70 ± 0.05 Bab | 39.40 ± 5.39 Bab |
|  | 30–50 | 4.95 ± 0.02 Bb | 12.63 ± 0.63 Da | 0.77 ± 0.05 Dab | 0.25 ± 0.01 Babc | 9.68 ± 0.16 Abc | 57.61 ± 5.83 Cab | 0.45 ± 0.10 Bab | 37.31 ± 6.93 Ba |

Notes: Uppercase letters (A, B, C, D) indicate different soil layers, and lowercase letters (a, b, c, d) indicate different cutting widths. TN (total nitrogen); TP (total phosphorus); TK (total potassium); HN (hydrolytic nitrogen); AP (available phosphorus); AK (available potassium).

In summary, when the cutting width was 8 m, there was no significant difference between the nutrient elements and the control group, and the total potassium and available potassium contents were higher than those in the control group. Therefore, the cutting width had abundant nutrients.

### 3.3. Relationships between Stand Characteristics and Soil Nutrients

As shown in Table 3, DBH was significantly negatively correlated with PH in the 0–10 cm and 20–30 cm soil layers, positively correlated with TK in the 0–10 cm and 30–50 cm soil layers, and positively correlated with AK in the 20–30 cm soil layer. DBS was positively correlated with PH and AP in the 10–20, 20–30, and 30–50 cm soil layers, with TN and HN in the 0–10 and 10–20 cm soil layers, and with TP in different soil layers. NB was significantly positively correlated with PH in the 20–30 cm soil layer. Our results showed that for DBS, the NB was significantly positively correlated with TK and HN in the 10–20 cm soil layer. There was no significant correlation between the other stand characteristics and the soil nutrient content.

**Table 3.** Correlations between stand characteristics and soil nutrients in moso bamboo.

|  | Soil Layers | DBH | DBS | NB | DBS: NB |
|---|---|---|---|---|---|
| PH | 0–10 | −0.298 | 0.163 | 0.001 | 0.287 |
|  | 10–20 | −0.542 ** | 0.484 ** | 0.143 | 0.223 |
|  | 20–30 | −0.409 * | 0.545 ** | 0.42 * | −0.053 |
|  | 30–50 | −0.248 | 0.523 ** | 0.258 | −0.055 |
| SOC | 0–10 | −0.063 | 0.366 | −0.041 | 0.183 |
|  | 10–20 | −0.088 | 0.436 | −0.065 | 0.178 |
|  | 20–30 | 0.089 | 0.211 | −0.116 | 0.128 |
|  | 30–50 | 0.083 | 0.234 | −0.229 | 0.245 |
| TN | 0–10 | −0.051 | 0.441 * | −0.054 | 0.274 |
|  | 10–20 | −0.036 | 0.441 * | −0.101 | 0.243 |
|  | 20–30 | 0.087 | 0.259 | −0.069 | 0.099 |
|  | 30–50 | 0.316 | 0.053 | −0.032 | 0.021 |
| TP | 0–10 | −0.236 | 0.843 ** | 0.148 | 0.264 |
|  | 10–20 | −0.143 | 0.801 ** | 0.12 | 0.187 |
|  | 20–30 | −0.094 | 0.796 ** | 0.006 | 0.297 |
|  | 30–50 | −0.224 | 0.505 ** | 0.043 | 0.148 |
| TK | 0–10 | 0.448 * | −0.355 | −0.223 | −0.203 |
|  | 10–20 | 0.17 | −0.189 | 0.071 | −0.443 * |
|  | 20–30 | 0.018 | −0.204 | −0.06 | −0.154 |
|  | 30–50 | 0.456 * | −0.296 | −0.149 | −0.287 |
| HN | 0–10 | −0.081 | 0.444 * | −0.103 | 0.362 |
|  | 10–20 | −0.336 | 0.514 * | 0.089 | 0.417 * |
|  | 20–30 | 0.125 | 0.199 | −0.026 | 0.07 |
|  | 30–50 | −0.135 | 0.284 | 0.033 | 0.114 |
| AP | 0–10 | −0.093 | 0.334 | −0.05 | 0.155 |
|  | 10–20 | −0.152 | 0.531 ** | 0.249 | −0.001 |
|  | 20–30 | −0.258 | 0.464 * | 0.186 | 0.192 |
|  | 30–50 | −0.302 | 0.545 ** | 0.167 | 0.277 |
| AK | 0–10 | 0.431 | −0.433 * | −0.37 | 0.011 |
|  | 10–20 | 0.396 | −0.171 | −0.169 | 0.001 |
|  | 20–30 | 0.619 ** | −0.412 * | −0.304 | −0.188 |
|  | 30–50 | 0.291 | −0.593 ** | −0.054 | −0.195 |

Note: SOC: soil organic carbon; TN: total nitrogen; TP: total phosphorus; TK: total potassium; HN: hydrolyzable nitrogen; AP: available phosphorus; AK: available potassium; DBS: degraded bamboo shoot; NB: number of new bamboo shoots (* $p < 0.01$; ** $p < 0.05$).

The first two principal components, component 1 (PC1) and component 2 (PC2), explained 45.53% and 20.98% of the variations, respectively (Figure 3). PCA analysis showed that stand characteristics (DBH and NB) were positively associated with TP and AP but negatively correlated with AK, TK, and SOC.

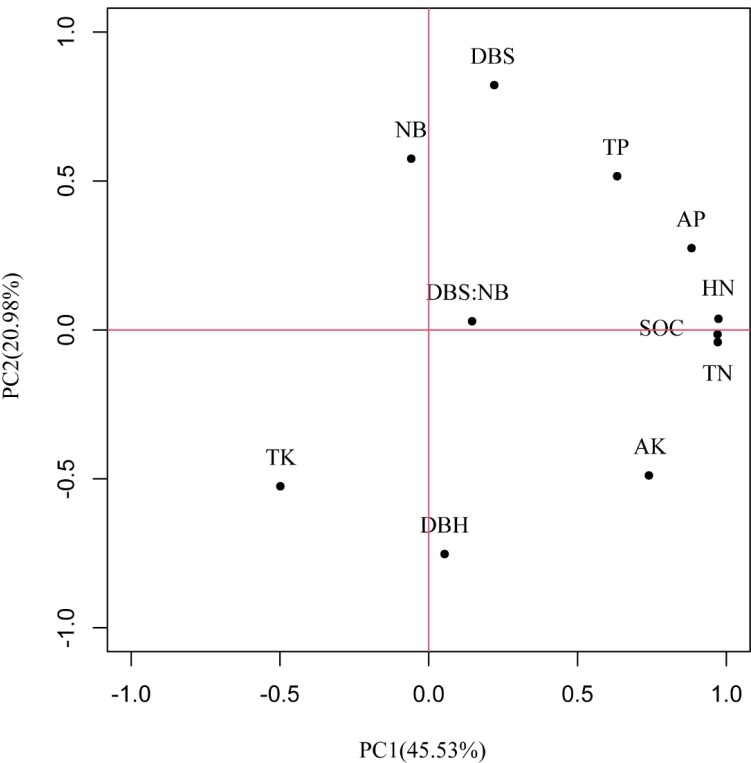

**Figure 3.** Principal component analysis (PCA) of soil nutrient content and stand characteristics.

### 3.4. Effects of Strip Cutting on Stand Characteristics of Moso Bamboo

For AAG (annual aboveground biomass of new bamboo shoots), there was no significant difference between the cutting and reserved zones with different widths ($p > 0.05$). However, for NBS (new bamboo shoots) and DBS (degraded bamboo shoots), there were significant differences among the bandwidths that displayed the same trend, that is, with an increase in bandwidth, the NBS and DBS increased (Table 4). When the cutting width was 8 m, there was more NBS and less DBS, and the AAG was relatively large. Thus, when the cutting width was 8 m, the bamboo forest exhibited better stand characteristics.

**Table 4.** Compare the stand characteristics of bamboo forest with different strip cutting and reserve zone.

| Measures | Cutting Width | AAG (kg/ha) | NBS (culm/ha) | DBS (culm/ha) |
|---|---|---|---|---|
| | 3 | 30.89 ± 7.17 a | 1491 ± 473 c | 281 ± 109 b |
| | 5 | 35.78 ± 12.15 a | 2164 ± 869 c | 291 ± 193 ab |
| C + R | 8 | 39.01 ± 9.59 a | 2161 ± 549 c | 295 ± 109 ab |
| | 12 | 32.89 ± 7.81 a | 2017 ± 582 b | 867 ± 100 ab |
| | 15 | 39.59 ± 2.52 a | 2507 ± 418 a | 1660 ± 237 a |
| | 3 | 29.56 ± 7.93 a | 1433 ± 544 c | 300 ± 125 b |
| | 5 | 37.19 ± 10.29 a | 2367 ± 816 c | 317 ± 195 ab |
| C | 8 | 31.61 ± 11.99 a | 2234 ± 629 c | 266 ± 102 ab |
| | 12 | 37.02 ± 7.65 a | 2403 ± 437 b | 861 ± 109 ab |
| | 15 | 41.10 ± 15.48 a | 2789 ± 245 a | 1700 ± 288 a |
| | 3 | 32.21 ± 6.04 a | 1556 ± 369 c | 259 ± 83 a |
| | 5 | 34.10 ± 13.88 a | 1920 ± 868 c | 260 ± 185 a |
| R | 8 | 40.99 ± 2.86 a | 2063 ± 399 c | 333 ± 106 a |
| | 12 | 26.70 ± 0.95 a | 1438 ± 21 b | 875 ± 83 a |
| | 15 | 37.32 ± 1.91 a | 2083 ± 217 a | 1600 ± 100 a |

Notes: C represents strip cutting measures, specifically strip clear cutting. R only performs selective cutting of mature bamboo. AAG: annual aboveground biomass of new bamboo shoots; NBS: new bamboo shoots; DBS: degraded bamboo shoots. Data are presented as the mean ± SD. Different lowercase letters (a, b, c) indicate significant differences among different measures ($p < 0.05$).

## 4. Discussion

Forest cutting is not only an important method for harvesting forest products but also an important means of forest regeneration. Cutting may lead to changes in the structure and quantity of the forest plant population [21,22]. This affects the process and the effect of the recovery and update. As a typical clonal plant, bamboo plants mainly germinate through the expansion of shoot buds on the underground bamboo whip to form new plants that reproduce and complete the self-renewal of the population [23]. Strip-cutting management involves cutting off all bamboo at a certain forestland width. Cutting removes a large number of aboveground trees, resulting in sharp changes in water, heat, light, and other forest conditions. It accelerates the decomposition rate of forest litter and cutting residues. Cutting increases the content of soil organic matter, total and effective contents of soil elements, such as N, P, and K, and soil enzyme activity [23], thereby improving forest soil quality. To some extent, the quantitative characteristics of degraded bamboo shoots and the number of new bamboo shoots can characterize the reproduction and regeneration abilities of moso bamboo forests. Analyzing these differences is very important to study the restoration of bamboo forests after strip cutting. In this study, after strip cutting of different widths, the number of Hsinchu and bamboo shoots were greater than that of the control group. With an increase in cutting width, the number of new bamboo in this study showed a trend of repeated increases and decreases, and the cutting width was the greatest in M12, which was higher than that in the CK group (Table 1). This study also found that cutting increased the soil nutrient content (TP, AP, TK, and AK) (Table 2). These results are similar to those previously reported [21–23,26]. Principal component analysis revealed that DBH and NB were positively associated with TP and AP but negatively correlated with AK, TK, and SOC. We concluded that the decrease in soil AK and TK contents (or increase in soil TP and AP contents) drives the increase in DBH and NB. This may be an example of the dynamic equilibrium theory of ecological chemometrics [15]. Cutting disturbance appears to disrupt the stability and integrity of bamboo forest ecosystems; however, timely and appropriate cutting management obtains bamboo and promotes the circulation of bamboo forest ecosystems and dynamic forest succession and renewal. Cutting can optimize plantation structure and restore forest ecological function in a short period [23,27,28], as well as keep nutrient elements stable and achieve a dynamic balance. It is very important to understand the characteristics of bamboo forests because these determine their structure. Taking reasonable management measures can adjust the competitive relationship between individuals of moso bamboo, providing a suitable environment to promote growth and development of moso bamboo and improve the quality of forest resources.

The higher SD value of the standard deviation in Table 1 may be due to the influence of the distribution of mother bamboo in the original reserved zone on the location of shoot emergence and withdrawal, which leads to a large difference in the number of shoots emerging and withdrawing under the same site conditions.

In this study, we consider the short-term effects of strip cutting. With an increase in cutting width, the degraded bamboo shoot of banded cutting first decreased and then increased, and the cutting width reached a minimum in M8, which is consistent with the results of a previous study [20–22].

After strip cutting, the competitive pressure was reduced, and the new bamboo received sufficient growth and nutritional space. Bamboo stalks have no cambium, and thickening of bamboo stalks does not require a periodic radial growth process. The DBH of Hsinchu was fixed during the shoot stage [29]. Therefore, the recovery of DBH of new bamboo after bamboo forest cutting can be used to evaluate the recovery quality of harvested bamboo forests to a certain extent. Different cutting methods have different effects on the DBH of post-cut new bamboo. Selective cutting may increase the average DBH of new bamboo the following year, but clear cutting usually leads to a decrease in the average DBH of new bamboo the following year [21,22,29]. In this study, the DBH of the new bamboo shoots decreased as the strip-cutting width increased (Table 1). As typical clonal plants, bamboo plants mainly germinate through the expansion of shoot

buds on underground bamboo whips to form new plants for reproduction and complete population self-renewal [30,31]. After strip cutting, the number of new bamboo plants increases, and their morphogenesis consumes substantial resources. After moso bamboo first encounters the survival of several clonal ramets and their subsequent growth and development, it reduces the DBH of new bamboo. When the cutting width was more than 8 m, the average DBH of the new bamboo increased significantly with the increase in cutting width. The growth of new bamboo at a large distance from the boundary of the cutting zone may be limited due to insufficient nutrient supply, resulting in a significant increase in the proportion of small- and medium-sized new bamboo [21,22,29]. Therefore, choosing a strip-cutting method with appropriate intensity is very important for the restoration of bamboo forests.

The results showed that the contents of organic carbon, total nitrogen, hydrolytic nitrogen, available phosphorus, and available potassium were the highest in the 0–10 cm soil layer. There are two predicted reasons for these results. First, under the influence of litter nutrient return and surrounding environmental factors, nutrients are first concentrated on the surface and then diffuse to a deeper level with water or other media [16]. Second, most animals and microorganisms live in the topsoil [32], which is conducive to the accumulation of nutrients in the topsoil and supports our findings. Compared with those in the CK group, the contents of SOC, TN, TP, AN, and AP in the banded cutting standard land increased, indicating that the release of soil nutrient elements occurred in a short time and that the nutrient content available for plants increased. Many soil nutrient contents in the 8-, 12-, and 15-m cutting zones were better than those in the 3- and 5-m cutting zones and CK, indicating that differences exist in the effects of strip-cutting intensity on the changes in soil chemical properties. In this study, SOC storage fluctuated with an increase in cutting width, and M8 and M15 were greater than in the other groups. This fluctuation may be the result of changes to the microclimate caused by cutting and the decomposition of forest floor C, which is temporarily stimulated as soils become warmer and possibly wetter due to reduced evapotranspiration [33]. These effects depended on the cutting width. Therefore, an appropriate cutting width may increase carbon reserves in the short term.

Forest cutting affects the annual average volume growth and aboveground biomass of the stand. With an increase in cutting intensity, the aboveground biomass of the stand shows a downward trend [34], and strip cutting has an impact on the average DBH and biomass increment per unit area of new bamboo. Compared with that in the reserved zone, the biomass per unit area of Hsinchu after cutting was reduced (Table 4), which is similar to the results obtained by Zeng et al. [21,22].

With the increase in cutting width, the number of new and degraded bamboo shoots showed an increasing trend. However, at the same cutting bandwidth, degraded bamboo shoots from the reserved zone were always smaller than those from the cutting zone. A potential reason could be that the nutrition of new bamboo shoots is mainly supplied by mother bamboo [30,31,35], and the mother bamboo is located in the reserve zone. Therefore, although it is easy to supply the reserve zone, the supply of the cutting zone is not timely, resulting in a large number of degraded bamboo shoots in the cutting zone. In this study, there was no significant difference in AAG between the reserved zone and the cutting zone with the same bandwidth, which may be due to the study focus on the biomass of the current year's new bamboo and the exclusion of the growth of the original bamboo forest in the reserved zone.

In summary, when the cutting width was 8 m, the number of bamboo shoots was higher, the number of degraded bamboo shoots was lower, and the amount of bamboo increased. There was no significant difference between the nutrient elements and the control group, and the total potassium and available potassium contents were higher than those in the control group. Therefore, the cutting width exhibited abundant nutrients suitable for cultivating bamboo.

## 5. Conclusions

The purpose of this experiment was to compare the changes in soil and stand characteristics of moso bamboo forests after strip cutting of different widths. The results showed that strip cutting had significant effects on degraded bamboo shoots, number of new bamboos, and their ratio. In addition, our study indicated that soil elements showed surface aggregation, and cutting increased the soil nutrient content to a certain extent. Stand characteristics (DBH and NB) were positively associated with TP and AP but negatively correlated with AK, TK, and SOC. When the cutting width was 8 m, the stand growth characteristics of the bamboo forest and the soil nutrient content were excellent. Soil carbon storage was also calculated. These results further confirm the effectiveness of strip cutting. These findings provide theoretical guidance for the formulation of scientific and sustainable strip-cutting methods for bamboo forests.

**Author Contributions:** X.Z. (Xiao Zhou), X.Z. (Xuan Zhang), C.L., Y.Z. and F.G. collected data; X.Z. (Xiao Zhou), X.Z. (Xuan Zhang), C.L., Y.Z. and F.G. analyzed data; X.Z. and F.G. wrote the manuscript, contributed critically to improve the manuscript, and gave final approval for publication. All authors have read and agreed to the published version of the manuscript.

**Funding:** This research was supported by the basic scientific research funding of International Center for Bamboo and Rattan (Grant No. 1632021003).

**Institutional Review Board Statement:** Not applicable.

**Informed Consent Statement:** Not applicable.

**Acknowledgments:** We would like to thank the basic scientific research funding of International Center for Bamboo and Rattan (Grant No. 1632021003) for the financial support of this study. We are thankful to two anonymous reviewers for the insightful comments and suggestions that helped improve the article.

**Conflicts of Interest:** The authors declare no conflict of interest.

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
