# Peer review of "Response of Moso Bamboo Growth and Soil Nutrient Content to Strip Cutting"

_forests, doi:10.3390/f13081293_

Round 1

Reviewer 1 Report

Title: Response of Moso bamboo growth and soil nutrient content to strip cutting                         

Authors: Xiao Zhou, Fengying Guan, Xuan Zhang, Chengji Li, Yang Zhou

To

The Editor in Chief

Journal of Forests

Dear Sir/Madam,

Thanks for giving me an opportunity to review the manuscript. I read manuscript now thoroughly and submitting my comments for your kind consideration. The details of comments are as follows:

Comments

Abstract:

·         In this section, the finding of the study is missing. Therefore, important findings of the study to be given.

 Introduction:

·         This section is written well but lacks of recent citation of current studies.

·         If possible, please provide hypothesis and clear objectives of the study.

Materials and Methods

·         Latitude and longitude of the study area to be given in the Location map.

·         Remaining parts of the materials and methods can be improved with important information’s only.

Results

·         This section is written well. However, can be improved.

Discussion:

·         The discussion section to be improved by providing additional recent citation of other relevant studies.

Conclusion:

·         This section to be describe in detailed. This section, needs to be revised with recent update of results and suggestions.

·         Including above the recent citation of current study to be added.

I personally feel that the authors have carried out good study and can be consider for publication. However, the suggested comments are essential before it final consideration for publication and quality of the Journal.

Author Response

Comments and Suggestions for Authors

Title: Response of Moso bamboo growth and soil nutrient content to strip cutting                         

Authors: Xiao Zhou, Fengying Guan, Xuan Zhang, Chengji Li, Yang Zhou

To

The Editor in Chief

Journal of Forests

Dear Sir/Madam,

Thanks for giving me an opportunity to review the manuscript. I read manuscript now thoroughly and submitting my comments for your kind consideration. The details of comments are as follows:

Comments

Abstract:

  • In this section, the finding of the study is missing. Therefore, important findings of the study to be given.

Response: thanks for your advice. We modified this in the manuscript.

 See line 188-22

Principal component analysis showed that stand characteristics (diameter at breast height and number of new bamboo shoots) were positively associated with total phosphorus and available phosphorus but negatively correlated with available potassium, total potassium, and soil organic carbon. A cutting width of 8 m resulted in rich nutrient content, which is suitable for bamboo cultivation.

 Introduction:

  • This section is written well but lacks of recent citation of current studies.

Response: thanks for your advice. We modified this in the manuscript.

See line 61-64

Many researchers choose the optimal cutting width by evaluating various indicators of different cutting widths [20,21,22,23]. Su et al. [20] selected the optimal width by comparing the stand characteristics of new bamboo after cutting. Zeng et al. [21,22] compared different soil nutrients, and Wang [23] compared the selection of microbial changes in different cutting widths.

  • If possible, please provide hypothesis and clear objectives of the study.

Response: thanks for your advice. We modified this in the manuscript.

See line 76-84

Materials and Methods

  • Latitude and longitude of the study area to be given in the Location map.

Response: thanks for your advice. We modified this in the manuscript.

See line 96-97

  • Remaining parts of the materials and methods can be improved with important information’s only.

Response: thanks for your advice. We modified this in the manuscript.

See line 87-167

Results

  • This section is written well. However, can be improved.

Response: thanks for your advice. We modified this in the manuscript.

See line 169-251

Discussion:

  • The discussion section to be improved by providing additional recent citation of other relevant studies.

Response: thanks for your advice. We modified this in the manuscript.

See line 267-356

Conclusion:

  • This section to be describe in detailed. This section, needs to be revised with recent update of results and suggestions.

Response: thanks for your advice. We modified this in the manuscript.

See line 375-385

  • Including above the recent citation of current study to be added.

Response: thanks for your advice. We modified this in the manuscript.

I personally feel that the authors have carried out good study and can be consider for publication. However, the suggested comments are essential before it final consideration for publication and quality of the Journal.

Reviewer 2 Report

Dear editor,

I have revised the ms forests-1823281 intitled “Response of Moso bamboo growth and soil nutrient content to strip cutting”. It is an interesting manuscript. Authors have evaluated the influence of strip cutting on the growth of Moso bamboo forests, soil nutrients, and litter content. 

Specific comments:

Abstract:

L17-22: Please provide numbers. An abstract it is not a menu. Here, we must associate number with the main results.

Introduction:

L28: References number 1 to 5 need to be updated.

L34-41: Provide numbers that show the economic value of the bamboo species.

L38: Reference number 6 needs to be updated.

L39: References number 9 to 12 need to be updated.

L44 and L48: Reference number 19 needs to be updated.

L48-49: References number 22 and 23 need to be updated.

L64: References number 28 and 29 need to be updated.

What was the main hypothesis of this study? Did the authors test any theory here? What was their main expectations about the study?

Material and Methods

L83: Improve Fig. 1 by adding details about the experimental design. There is no need to have separate figures, such as Fig. 1 and 2. They must be presented in just one figure.

L141-142: Statistical analyses must be improved. Did the authors test data normality and homogeneity of variance? Why didn’t the authors use a correlation analysis? It could be interesting add a PCA analysis here. What about the ANOVA type? Where did the authors use an one-way ANOVA and a two-way ANOVA?

Results

 I did not find the litter contents that were described in the aims of this study.

L159 Please improve the Table 1 presentation. Write a full title here.

Could the authors explain the high values of standard deviation into table 1?

L189 Please improve the Figure 3 presentation. Write a full title here.

L194 Please improve the Table 2 presentation. Write a full title here.

Discussion

Some references must be updated into the discussion section (e.g., references number 32, 37, 43,44, and 45).

Did the results support the main hypothesis of this study?

This section must be rewritten to follow the recommended correlation an PCA analysis.

Author Response

Comments and Suggestions for Authors

Dear editor,

I have revised the ms forests-1823281 intitled “Response of Moso bamboo growth and soil nutrient content to strip cutting”. It is an interesting manuscript. Authors have evaluated the influence of strip cutting on the growth of Moso bamboo forests, soil nutrients, and litter content. 

Specific comments:

Abstract:

L17-22: Please provide numbers. An abstract it is not a menu. Here, we must associate number with the main results.

Response: thanks for your advice. We modified this in the manuscript.

See line 18-22

Principal component analysis showed that stand characteristics (diameter at breast height and number of new bamboo shoots) were positively associated with total phosphorus and available phosphorus but negatively correlated with available potassium, total potassium, and soil organic carbon. A cutting width of 8 m resulted in rich nutrient content, which is suitable for bamboo cultivation.

Introduction:

L28: References number 1 to 5 need to be updated.

Response: thanks for your advice. We modified this in the manuscript.

L34-41: Provide numbers that show the economic value of the bamboo species.

Response: thanks for your advice. We modified this in the manuscript.

 See line 37-39

The annual biomass of bamboo stalk in moso bamboo forest can generally reach 8.25–9 t/ ha, with an input–output ratio of 1:2–1:4, which can increase the economic benefit 2–3 times more than with the management of general timber forest species [6].

L38: Reference number 6 needs to be updated.

Response: thanks for your advice. We modified this in the manuscript.

L39: References number 9 to 12 need to be updated.

Response: thanks for your advice. We modified this in the manuscript.

L44 and L48: Reference number 19 needs to be updated.

Response: thanks for your advice. We modified this in the manuscript.

L48-49: References number 22 and 23 need to be updated.

Response: thanks for your advice. We modified this in the manuscript.

L64: References number 28 and 29 need to be updated.

Response: thanks for your advice. We modified this in the manuscript.

What was the main hypothesis of this study? Did the authors test any theory here? What was their main expectations about the study?

Response: thanks for your advice. We modified this in the manuscript.

See line 76-84

The effects of different cutting widths on soil nutrients and stand characteristics of bamboo forests are unclear. In this study, strip cutting with different cutting widths was performed using the unrecovered sample plot as the control. It is assumed that banded cutting can adjust the competitive relationship between moso bamboo individuals, provide a suitable environment for moso bamboo growth, and thus promote growth and development. This study determined (1) whether there are differences in soil nutrient content and stand characteristics at different cutting widths, (2) the effects of soil nutrient factors on stand characteristics after harvesting at different cutting widths, and (3) the optimal cutting width by correlation analysis.

Material and Methods

L83: Improve Fig. 1 by adding details about the experimental design. There is no need to have separate figures, such as Fig. 1 and 2. They must be presented in just one figure.

Response: thanks for your advice. We modified this in the manuscript.

See line 96-97

L141-142: Statistical analyses must be improved. Did the authors test data normality and homogeneity of variance? Why didn’t the authors use a correlation analysis? It could be interesting add a PCA analysis here. What about the ANOVA type? Where did the authors use an one-way ANOVA and a two-way ANOVA?

Response: thanks for your advice. We modified this in the manuscript.

 See line 159-167, 231-251

Results

  I did not find the litter contents that were described in the aims of this study.

Response: thanks for your advice. We modified this in the manuscript. See line

For the reason of expression, I have deleted the expression of litter in the introduction

L159 Please improve the Table 1 presentation. Write a full title here.

Response: thanks for your advice. We modified this in the manuscript.

See line 184-188

Could the authors explain the high values of standard deviation into table 1?

Response: thanks for your advice. We modified this in the manuscript.

See line 302-305

The higher SD value of the standard deviation in Table 1 may be due to the influence of the distribution of mother bamboo in the original reserved zone on the location of shoot emergence and withdrawal, which leads to a large difference in the number of shoots emerging and withdrawing under the same site conditions.

L189 Please improve the Figure 3 presentation. Write a full title here.

Response: thanks for your advice. We modified this in the manuscript.

See line 219-220

L194 Please improve the Table 2 presentation. Write a full title here.

Response: thanks for your advice. We modified this in the manuscript.

See line 211-216

Discussion

Some references must be updated into the discussion section (e.g., references number 32, 37, 43,44, and 45).

Did the results support the main hypothesis of this study?

Response: thanks for your advice. We modified this in the manuscript.

See line 267-301

This section must be rewritten to follow the recommended correlation an PCA analysis.

Response: thanks for your advice. We modified this in the manuscript.

See line 267-301

Round 2

Reviewer 1 Report

Based on my previous review report, I observed that authors have made substantial changes in the manuscript. I recommend the manuscript for publication in the Journal.

Reviewer 2 Report

Dear editor,

I have revised the manuscript id forest-1823281. All my comments in the previously version were responded accordingly. 

The manuscript is now suitable for publication in Forests MDPI.